# Age and Tumor Stage Interplay in Intrahepatic Cholangiocarcinoma: Prognostic Factors, Mortality Trends, and Therapeutic Implications from a SEER-Based Analysis

**DOI:** 10.3390/diseases13020031

**Published:** 2025-01-25

**Authors:** Ayrton Bangolo, Vignesh K. Nagesh, Hadrian Hoang-Vu Tran, Brooke Sens, Daniel Elias, Behzad Amoozgar, Chase Tomasino, Izage Kianifar Aguilar, Charlene Mansour, Elizabeth Gagen, Lili Zhang, Sarvarinder Gill, Nisrene Jebara, Emma Madigan, Christin Candela, Dohaa Amin, Peter Giunta, Shubhangi Singh, Aman Siddiqui, Auda Auda, Paul Peej, Timophyll Y. H. Fong, Simcha Weissman, Printhia Matshi Lihau, John Bukasa-Kakamba

**Affiliations:** 1Department of Hematology and Oncology, John Theurer Cancer Center, Hackensack University Medical Center, Hackensack, NJ 07601, USA; behzad.amoozgar@hmhn.org (B.A.); lili.zhang@hmhn.org (L.Z.); sarvarinder.gill@hmhn.org (S.G.); 2Department of Internal Medicine, Hackensack Palisades Medical Center, North Bergen, NJ 07047, USA; tranhadrian@gmail.com (H.H.-V.T.); izagekianifar@gmail.com (I.K.A.); simchaweissman@gmail.com (S.W.); 3Rutgers New Jersey Medical School, Newark, NJ 07103, USA; de217@njms.rutgers.edu (D.E.); cm1621@njms.rutgers.edu (C.M.); 4Columbia University School of Nursing, New York, NY 10032, USA; nj2454@cumc.columbia.edu; 5Department of Family Medicine, Hackensack Palisades Medical Center, North Bergen, NJ 07047, USA; amanksiddiqui1@gmail.com (A.S.); audaxyz@gmail.com (A.A.); 6Department of Research and Innovation, Congolese National Cancer Control Center, Kinshasa, Democratic Republic of the Congo; printhialihau@yahoo.fr; 7Department of Endocrinology and Nuclear Medicine, Kinshasa University Clinics, Kinshasa, Democratic Republic of the Congo; johnbukasa73@gmail.com

**Keywords:** age-related prognosis, tumor stage, intrahepatic cholangiocarcinoma, SEER database, survival outcomes, therapeutic interventions, mortality trends

## Abstract

Background: Intrahepatic cholangiocarcinoma (ICC), a malignancy originating from the epithelial cells of bile ducts, has shown a notable rise in its incidence over the years. It ranks as the second most frequent primary liver cancer after hepatocellular carcinoma. This study investigates how independent prognostic factors, specifically, age and tumor stage, interact to impact mortality in ICC patients. Furthermore, it examines the clinical features, survival rates, and prognostic indicators of ICC cases diagnosed between 2010 and 2017. Methods: Using data from 5083 patients obtained from the Surveillance, Epidemiology, and End Results (SEER) database, this study evaluated demographic and clinical factors alongside overall mortality (OM) and cancer-specific mortality (CSM). Variables achieving a *p*-value below 0.1 in univariate Cox regression analysis were incorporated into multivariate Cox regression models to identify independent prognostic factors. Hazard ratios (HRs) exceeding 1 were interpreted as markers of poor prognosis. Additionally, this study explored the interaction between age and tumor stage in shaping survival outcomes. Results: The multivariate Cox proportional hazards analysis indicated higher OM in males (HR = 1.19, 95% CI: 1.12–1.26, *p* < 0.01) and residents of metropolitan counties with populations exceeding 250,000 (HR = 1.15, 95% CI: 1.01–1.31, *p* < 0.05). Conversely, lower OM was observed in individuals aged 40–59 years (HR = 0.58, 95% CI: 0.38–0.89, *p* < 0.05), those aged 60–79 years (HR = 0.65, 95% CI: 0.43–0.98, *p* < 0.05), and patients who received radiation therapy (HR = 0.78, 95% CI: 0.72–0.85, *p* < 0.01), chemotherapy (HR = 0.54, 95% CI: 0.51–0.58, *p* < 0.01), or surgery (HR = 0.29, 95% CI: 0.26–0.31, *p* < 0.01). For CSM, males exhibited higher risks (HR = 1.17, 95% CI: 1.10–1.25, *p* < 0.01), as did individuals in metropolitan counties with populations over 250,000 (HR = 1.18, 95% CI: 1.03–1.35, *p* < 0.05). Reduced CSM was observed in patients aged 40–59 years (HR = 0.52, 95% CI: 0.34–0.79, *p* < 0.01), those aged 60–79 years (HR = 0.57, 95% CI: 0.38–0.86, *p* < 0.01), and those undergoing radiation therapy (HR = 0.76, 95% CI: 0.70–0.83, *p* < 0.01), chemotherapy (HR = 0.55, 95% CI: 0.51–0.59, *p* < 0.01), or surgery (HR = 0.27, 95% CI: 0.25–0.30, *p* < 0.01). When examining the interaction between age and tumor stage, higher OM was observed in patients aged 40–59 with tumors involving lymph nodes (HR = 1.26, 95% CI: 1.14–2.67, *p* < 0.05). Similarly, CSM was elevated in patients aged 40–59 with lymph node involvement alone (HR = 2.60, 95% CI: 1.26–5.36, *p* < 0.05) or with direct spread (HR = 2.81, 95% CI: 1.04–7.61, *p* < 0.05). Among those aged 60–79, higher CSM was noted in cases with lymph node involvement only (HR = 2.24, 95% CI: 1.11–4.50, *p* < 0.05) or lymph node involvement accompanied by direct extension (HR = 2.93, 95% CI: 1.10–7.82, *p* < 0.05). Conclusions: This retrospective analysis, utilizing data from the SEER database, provides new insights into mortality patterns in intrahepatic cholangiocarcinoma (ICC). This study identifies a significant interplay between two key prognostic factors, emphasizing their collective role in influencing mortality outcomes. Despite the predominance of advanced-stage diagnoses, our analysis underscores the substantial survival benefits associated with treatment interventions, with surgical procedures demonstrating the most pronounced impact. These findings highlight the importance of recognizing patients who may benefit from timely and intensive therapeutic strategies. Furthermore, the results underscore the need for future prospective randomized studies to deepen our understanding of these interactions in ICC, particularly as advancements in precision oncology continue to refine patient care.

## 1. Introduction

Intrahepatic cholangiocarcinoma (ICC) arises from the epithelial cells within the intrahepatic and extrahepatic bile ducts, situated near the segmental biliary ducts [1]. It is the second most common primary liver malignancy after hepatocellular carcinoma, with a steadily increasing global incidence [2]. Currently, ICC accounts for approximately 10% of all cholangiocarcinomas and has experienced a dramatic rise in incidence—over 140%—over the past four decades [3,4]. Its current occurrence rate is estimated at 1–2 cases per 100,000 individuals [3].

The clinical presentation of ICC is often vague, with nonspecific symptoms such as abdominal discomfort, weight loss, and general malaise. These subtle manifestations frequently result in delayed diagnoses and poorer prognoses [5,6]. Compared to cholangiocarcinomas in other locations, ICC patients are less likely to exhibit jaundice. Instead, they may present with dull pain in the right upper quadrant, unexplained weight loss, elevated alkaline phosphatase levels, or, less commonly, fever. Notably, many cases remain asymptomatic and are discovered incidentally through imaging studies [7].

Established risk factors for cholangiocarcinoma (CCA), such as hepatobiliary flukes, primary sclerosing cholangitis (PSC), biliary tract cysts, hepatolithiasis, and toxin exposure, contribute to chronic biliary inflammation and increased cellular turnover. Additionally, risk factors for ICC overlap with those for hepatocellular carcinoma (HCC), including liver cirrhosis, chronic hepatitis B and C, obesity, diabetes, and excessive alcohol consumption [8,9].

At the time of diagnosis, more than 75% of patients with intrahepatic cholangiocarcinoma (ICC) are aged 65 years or older [3]. The disease is associated with a high mortality rate, with a 5-year overall survival rate of less than 10% [10]. Diagnosis typically relies on imaging studies and tissue biopsy, complemented increasingly by laboratory evaluations that include liver function tests and tumor markers [9,11]. A core-needle biopsy allows for the final diagnosis of ICC as well as the opportunity to analyze the tumor histopathologically and pathogenetically [10]. Treatment options for patients are diverse and often involve a multimodal approach, including surgical resection, perioperative chemotherapy, liver-directed therapies, transplantation, and systemic treatments such as cytotoxic agents, targeted therapies, and immunotherapy [9,12].

While previous research has explored the individual roles of age [13] and tumor stage [14] in shaping survival outcomes for patients with intrahepatic cholangiocarcinoma (ICC), the combined impact of these factors remains unexamined. To bridge this gap, we undertook a comprehensive retrospective cohort analysis using data from the Surveillance, Epidemiology, and End Results (SEER) database, focusing on ICC cases diagnosed between 2010 and 2017 to evaluate the interplay between age and tumor stage on survival outcomes. We hope this knowledge enhances our understanding of the multifaceted factors affecting ICC prognosis, paving the way for more targeted and informed approaches in the clinical management of this complex disease.

## 2. Methods

This retrospective cohort study utilized data from the SEER Research Plus database, which includes 18 population-based registries, and the November 2020 submission (http://www.seer.cancer.gov), accessed on 24 September 2023. Sponsored by the United States National Cancer Institute (US NCI), the SEER Program is one of the most extensive and detailed resources for cancer-related data in the U.S. The SEER 18 database provides comprehensive information on cancer incidence, clinicopathological features, and survival outcomes, covering approximately 28% of the U.S. population [5].

Data for this study were extracted from the SEER database using histological codes specific to intrahepatic cholangiocarcinoma (ICC) [5]. Patients with incomplete data on age at diagnosis, race, or tumor stage were excluded from the analysis. This study focused on selected variables as the primary exposures. Overall mortality (OM) was defined as death from any cause during this study period, whereas cancer-specific mortality (CSM) referred to deaths directly related to complications arising from ICC.

The dataset included a range of variables, such as patient age at diagnosis, gender, racial classification (White, Black, or other), ethnicity (Hispanic or non-Hispanic), primary tumor location, tumor stage at diagnosis (categorized as localized, regional, or distant), geographic region, annual income, marital status, year of diagnosis, and treatment approaches, including surgery, radiation therapy, and chemotherapy.

The Cox proportional hazards regression model assumes a consistent proportionality of hazard rates over time. Variables with a *p*-value less than 0.1 in the univariate analysis were advanced to the multivariate model to determine independent predictors of overall mortality (OM) and cancer-specific mortality (CSM). A hazard ratio (HR) greater than 1 was indicative of a negative prognostic factor. Statistical analyses were conducted as two-sided, employing a 95% confidence interval, with *p*-values below 0.01 considered statistically significant. All computations were carried out using STATA version 18. Additionally, the interaction between stage at diagnosis and histology type was examined, with HR > 1 indicating adverse outcomes.

## 3. Results

A total of 5803 cases of intrahepatic cholangiocarcinoma (ICC) diagnosed between 2010 and 2017 were included in this analysis. Table 1 outlines the demographic and clinicopathologic features of the patient cohort. The largest proportion of patients (57.66%) were between 60 and 79 years old, and a majority identified as non-Hispanic White (61.23%). Most patients lived in metropolitan areas with populations greater than 1 million (59.93%) and reported annual incomes of USD 75,000 or higher (39.76%). The age breakdown revealed that 2.69% of patients were under 40 years old, 26.56% were aged 40–59, 57.66% fell into the 60–79 age category, and 13.10% were 80 years or older. Marital status varied, with 57.32% married, 19.87% single, 10.43% divorced or separated, and 12.39% widowed.

The distribution of tumor stages showed that 43.72% of patients were diagnosed with distant-stage disease. Localized cases represented 24.42%, while 17.71% had regional disease with direct extension only, 7.55% had involvement of regional lymph nodes only, and 6.60% had both direct extension and lymph node involvement. In terms of racial composition, the cohort consisted of 61.23% non-Hispanic Whites, 17.03% Hispanics, 13.56% individuals of other racial backgrounds, and 8.19% non-Hispanic Blacks.

The majority of patients (59.93%) resided in metropolitan areas with populations exceeding 1 million. In terms of income, the largest proportion (39.76%) reported annual earnings of USD 75,000 or higher. Treatment trends revealed that 86.11% of patients did not receive radiation therapy, while 53.53% underwent chemotherapy, and 20.11% underwent surgical procedures. When examining the distribution of cases by year, diagnoses ranged from 8.63% in 2010 to 17.96% in 2017, averaging approximately 40 new cases annually over the study period.

We provide Table 2, which presents a crude analysis of factors associated with overall mortality and cancer-specific mortality in U.S. patients diagnosed with intrahepatic cholangiocarcinoma (ICC) between 2010 and 2017. This analysis evaluates mortality outcomes in relation to various patient characteristics. Men demonstrated a higher risk than women, with crude hazard ratios of 1.16 (95% CI: 1.10–1.23) for overall mortality and 1.15 (95% CI: 1.08–1.22) for cancer-specific mortality. Age at diagnosis significantly influenced outcomes, with older individuals facing greater mortality risks. In patients aged 80 years or older, the hazard ratio reached 1.98 (95% CI: 1.63–2.40) for overall mortality and 1.79 (95% CI: 1.47–2.18) for cancer-specific mortality. Widowed patients showed elevated risks, with hazard ratios of 1.20 (95% CI: 1.09–1.31) for overall mortality and 1.18 (95% CI: 1.07–1.30) for cancer-specific mortality. Advancing tumor stage was associated with progressively higher hazard ratios, culminating in distant-stage disease with hazard ratios of 2.64 (95% CI: 2.44–2.85) for overall mortality and 2.87 (95% CI: 2.64–3.12) for cancer-specific mortality.

Differences in mortality risks were evident across geographic regions, income levels, and treatment approaches, with statistically significant variations highlighted by *p*-values (* *p* < 0.05, ** *p* < 0.01). Among treatment modalities, surgery offered a pronounced protective benefit, with hazard ratios of 0.26 (95% CI: 0.24–0.28) for overall mortality and 0.25 (95% CI: 0.23–0.27) for cancer-specific mortality. These results highlight the critical role of demographic and clinicopathologic factors in shaping mortality outcomes for patients with intrahepatic cholangiocarcinoma.

Table 3 presents the results of multivariate Cox proportional hazards regression analyses, highlighting factors associated with overall mortality and cancer-specific mortality in U.S. patients diagnosed with intrahepatic cholangiocarcinoma (ICC) between 2010 and 2017. Male patients demonstrated a higher risk of overall mortality, with an adjusted hazard ratio (HR) of 1.19 (95% CI: 1.12–1.26, ** *p* < 0.01) compared to females. Age emerged as a significant determinant, with reduced risks observed in patients aged 40–59 years (HR: 0.58, 95% CI: 0.38–0.89, * *p* < 0.05) and 60–79 years (HR: 0.65, 95% CI: 0.43–0.98, * *p* < 0.05). In contrast, marital status showed no significant associations with overall mortality.

Regarding the tumor stage, the distant stage did not exhibit a significant impact on overall mortality. No notable associations were identified for race, place of residence, or annual income. However, male patients were found to have an increased risk of cancer-specific mortality, with an adjusted hazard ratio (HR) of 1.17 (95% CI: 1.10–1.25, ** *p* < 0.01).

Age continued to play a protective role, with the 40–59 age group showing reduced cancer-specific mortality (HR: 0.52, 95% CI: 0.34–0.79, ** *p* < 0.01), as did the 60–79 age group (HR: 0.57, 95% CI: 0.38–0.86, ** *p* < 0.01). Living in counties with populations around 250,000 was associated with increased cancer-related mortality (HR: 1.18, 95% CI: 1.03–1.35, ** *p* < 0.01). On the other hand, receiving chemotherapy (HR: 0.55, 95% CI: 0.51–0.59, ** *p* < 0.01) and undergoing surgery (HR: 0.27, 95% CI: 0.25–0.30, ** *p* < 0.01) were strongly linked to significantly reduced risks of cancer-specific mortality.

In conclusion, the multivariate analyses emphasize the intricate role of various factors in shaping both all-cause and cancer-specific mortality among patients with intrahepatic cholangiocarcinoma. These findings highlight the importance of considering multiple variables when evaluating survival outcomes.

The multivariate Cox proportional hazards regression analyses, examining the interaction between tumor stage and age in U.S. patients diagnosed with intrahepatic cholangiocarcinoma from 2010 to 2017 (Table 4), uncovered significant relationships between these variables and both overall mortality (OM) and cancer-specific mortality (CSM).

Notably, the 40–59 age group within stage III demonstrated a significantly increased hazard ratio for both all-cause mortality (HR = 2.30, 95% CI 1.14–4.67, * *p* < 0.05) and cancer-related mortality (HR = 2.60, 95% CI 1.26–5.36, * *p* < 0.05). Additionally, the 40–59 age group in stage V exhibited a substantially higher hazard ratio for both all-cause mortality (HR = 2.16, 95% CI 1.32–3.54, ** *p* < 0.01) and cancer-related mortality (HR = 2.36, 95% CI 1.44–3.88, ** *p* < 0.01). The findings suggest that the interaction between specific age groups and tumor stages significantly influences mortality outcomes in intrahepatic cholangiocarcinoma patients, emphasizing the importance of considering these joint effects for a more nuanced understanding of prognosis.

## 4. Discussion

In this large, retrospective SEER database study, the population consisted mostly of non-Hispanic whites, with patients diagnosed mostly after 60 years of age. The patients in our database consisted predominantly of people in the metropolitan areas. There was almost an equal representation of both genders in our study. The majority of the patients did not undergo surgery or radiation therapy. However, most patients received chemotherapy. The mortality was found to be higher in males and the elderly population. Advanced-stage tumors were associated with higher mortality; however, the patients who received therapies had a lower mortality rate than those who did not undergo any treatment. Furthermore, a unique result of our study is that there was a significant interaction between age and advanced tumor stages, which adversely affected the mortality of younger patients diagnosed with iCCA.

Intrahepatic cholangiocarcinoma is a very aggressive tumor and is mainly diagnosed in the age of 50–70, which mirrors the patient population in our study [13,15]. In the multivariate analysis, the mortality was higher in males compared to females, as highlighted in previous studies [13,15]. The exact reason for this variation of mortality in gender is unclear; however, it could be attributed to the fact that males have a higher incidence of primary sclerosing cholangitis, hepatitis C, and liver cirrhosis, as demonstrated in the previous literature, as these comorbid conditions increase the risk of cholangiocarcinoma [16,17,18,19]. In our study, we found that advanced age was associated with higher mortality. Studies conducted by Antwi et al. and Yao et al. also showed an increase in mortality of iCCA with age [20,21]. A possible cause for the higher mortality in the elderly population could be due to the side effects of treatment modalities causing malnutrition in the elderly, hindering them from pursuing further treatment [13]. The incidence and mortality of iCCA were found to be higher in metropolitan counties in our study. This could be due to the fact that metropolitan counties have better resources and access to healthcare, and the patients in rural settings would lack follow-up and, hence, the data on mortality could be limited.

iCCA is commonly diagnosed in the later stages of the tumor, however, if diagnosed at an early stage, the patient can undergo partial hepatectomy along with adjuvant chemotherapy, given the high risk of recurrence [21,22,23,24,25]. Chemotherapy with gemcitabine with or without cisplatin is the main modality of treatment in advanced iCCA. In our study, chemotherapy was shown to decrease mortality in our findings, which has been reflected in the studies conducted by Valle et al. and Ramirez et al. [26,27]. In our study, patients who underwent surgery showed a significant decrease in mortality. Studies have shown that anatomical resection with vascular reconstruction has shown to decrease mortality of iCCA [28,29,30]. External beam radiation therapy (EBRT) and selective-internal radiation therapy (SIRT) have proven to be beneficial in downstaging iCCA, and, eventually, patients undergo surgery [31].

In the univariate analyses, high-grade tumors proved to have higher mortality, and these findings are consistent with studies demonstrating the inoperability of advanced-stage iCCA leading to higher mortality [23,24]. Lymph node metastases are associated with very poor outcomes [25]. In the multivariate analyses, there was no significant mortality with higher tumor grade. A noteworthy finding in our study was the significant interaction observed between age and tumor stage. While age correlated with increased mortality, intriguingly, younger individuals with stage III and IV iCCA exhibited higher mortality rates compared to patients over 80 years old. This phenomenon could potentially stem from genomic variances leading to distinct clinical behaviors and resistance to chemotherapy [32]. Similar trends have been noted in colorectal carcinoma, where younger patients often present with signet ring histology and advanced-stage tumors [33,34]. This unique finding of iCCA in our study has not been studied. Our study revealed a significant interaction between age and tumor stage, which could pave the way for studies to elucidate genomic variations in the younger population predisposing them to aggressive cancer and conduct randomized control trials (RCTs) in developing novel therapies to treat iCCA. Studies should also be conducted to develop a screening technique for these patients to aid physicians in early diagnosis of these patients to provide aggressive therapy.

This study evaluated a substantial cohort of 5083 patients with histologically confirmed intrahepatic cholangiocarcinoma (iCCA), utilizing a carefully selected population based on strict inclusion and exclusion criteria. However, several limitations should be acknowledged. The SEER database lacks information on additional comorbidities and the details of surgical or chemotherapeutic treatments administered to patients. Additionally, data on Body Mass Index (BMI) and lifestyle factors are not captured in the database. Lastly, the retrospective nature of the study design may introduce inherent biases.

## 5. Conclusions

Our comprehensive analysis of 5803 US patients diagnosed with intrahepatic cholangiocarcinoma (iCCA) between 2010 and 2017 yielded statistically significant insights into the factors influencing mortality. In summary, we found that age is a crucial determinant, with patients aged 80 and above experiencing a nearly twofold increase in overall mortality compared to those aged 0–39. The impact of marital status was also notable, with widowed individuals facing an elevated risk of both overall and cancer-related mortality. The tumor stage exhibited a significant gradient effect, with each progressive stage correlating with higher mortality rates. For example, individuals diagnosed with intrahepatic cholangiocarcinoma at a distant stage exhibited a mortality risk more than twice that of those with localized cases. Placing our study in the context of the existing literature, our findings align with and extend previous research, providing robust statistical support to the multifactorial nature of iCCA prognosis. From a clinical perspective, the statistical significance of these factors underscores the need for personalized care. Clinicians should be attuned to age-specific considerations, the impact of marital status, and the critical role of tumor staging in prognosis. Future research directions should leverage these statistical insights to delve into the molecular underpinnings of iCCA, allowing for targeted therapeutic approaches. Additionally, initiatives aimed at reducing healthcare disparities and enhancing access to advanced treatments are imperative, with the goal of translating statistical associations into improved patient outcomes. In conclusion, our study, backed by robust statistical evidence, not only enhances the understanding of iCCA determinants but also provides a quantitative foundation for refining prognostic and therapeutic strategies in the management of intrahepatic cholangiocarcinoma.

## Figures and Tables

**Table 1 diseases-13-00031-t001:** Characteristics of U.S. patients with intrahepatic cholangiocarcinoma diagnosed from 2010 to 2017, including demographic and clinicopathologic data.

Characteristics		
**Total**	**N**	**%**
	5803	100
**Gender**		
Female	2867	49.41
Male	2936	50.59
**Age at which patient was diagnosed (years old)**		
Age ranges from 0 to 39	156	2.69
Age ranges from 40 to 59	1541	26.56
Age ranges from 60 to 79	3346	57.66
Age 80 years old and above	760	13.10
**Marital status**		
Married	3326	57.32
Single	1153	19.87
Divorced/separated	605	10.43
Widowed	719	12.39
**Tumor stage**		
Localized	1417	24.42
Regional by direct extension only	1028	17.71
Involvement of regional lymph nodes only	438	7.55
Both direct extension and lymph node involvement	383	6.60
Distant metastasis	2537	43.72
**Race**		
Non-Hispanic white	3553	61.23
Non-Hispanic black	475	8.19
Hispanic	988	17.03
Other	787	13.56
**Regions which patient lived in**		
Metropolitan counties with populations over 1 million	3478	59.93
metropolitan counties with populations between 250,000 and 1 million	1286	22.16
metropolitan counties with populations below 250,000	433	7.46
nonmetropolitan counties bordering a metropolitan area	353	6.08
nonmetropolitan counties without proximity to a metropolitan area	253	4.36
Annual income (or Income per year)		
Income below USD 35,000	47	0.81
Income between range of USD35,000 and 44,999	264	4.55
Income between range of USD 45,000 and 54,999	614	10.58
USD 55,000–64,999	1068	18.40
USD 65,000–74,999	1503	25.90
USD 75,000+	2307	39.76
**Radiation**		
No	4997	86.11
Yes	806	13.89
**Chemotherapy**		
No	2697	46.48
Yes	3106	53.53
**Surgery**		
No	4636	79.89
Yes	1167	20.11
**Year of diagnosis**		
2010	501	8.63
2011	514	8.86
2012	566	9.75
2013	654	11.27
2014	781	13.46
2015	853	14.70
2016	892	15.37
2017	1042	17.96

**Table 2 diseases-13-00031-t002:** Crude analysis of factors influencing overall mortality and cancer-specific mortality in U.S. patients diagnosed with intrahepatic cholangiocarcinoma from 2010 to 2017.

Characteristics	Overall Mortality	Cancer-Specific Mortality
**Gender**		
Female	1 (reference)	1 (reference)
Male	Increased risk: 1.16 (1.10–1.23) **	Higher risk: 1.15 (1.08–1.22) **
**Age at Diagnosis (years)**		
Age from 0 to 39	1 (reference)	1 (reference)
Age from 40 to 59	Slightly elevated: 1.13 (0.94–1.36)	Marginally higher: 1.10 (0.91–1.33)
Age from 60 to 79	Moderately increased: 1.27 (1.06–1.52) **	Elevated risk: 1.21 (1.01–1.45) *
Age 80 and above	Significantly higher: 1.98 (1.63–2.40) **	Markedly increased: 1.79 (1.47–2.18) **
**Marital Status**		
Married	1 (reference)	1 (reference)
Single	Similar risk: 1.00 (0.93–1.08)	Comparable outcome: 0.99 (0.91–1.07)
Divorced/Separated	Slightly elevated: 1.03 (0.93–1.13)	Marginally higher: 1.03 (0.93–1.14)
Widowed	Noticeably increased: 1.20 (1.09–1.31) **	Substantially elevated: 1.18 (1.07–1.30) **
**Tumor Stage**		
Localized	1 (reference)	1 (reference)
Regional (Direct Extension Only)	Moderately elevated: 1.63 (1.48–1.79) **	Higher risk: 1.72 (1.56–1.90) **
Regional (Lymph Nodes Only)	Markedly increased: 1.78 (1.58–2.01) **	Substantial risk: 1.92 (1.69–2.18) **
Regional (Direct Extension + LN)	Considerably elevated: 1.88 (1.66–2.13) **	Noticeably increased: 2.03 (1.79–2.31) **
Distant	Most significant: 2.64 (2.44–2.85) **	Highest risk: 2.87 (2.64–3.12) **
Race		
Non-Hispanic White	1 (reference)	1 (reference)
Non-Hispanic Black	Comparable risk: 1.03 (0.92–1.15)	Similar risk: 1.04 (0.93–1.16)
Hispanic	Minimal difference: 1.00 (0.93–1.09)	Negligible risk: 0.99 (0.91–1.08)
Other	Slightly elevated: 1.02 (0.93–1.11)	Marginally higher: 1.01 (0.92–1.10)
**Living Area (Counties)**		
Counties in metropolitan areas > 1 million	1 (reference)	1 (reference)
Counties in metropolitan areas (250k–1 million)	Modestly increased: 1.07 (0.99–1.15)	Slightly elevated: 1.08 (1.00–1.16)
Counties in metropolitan areas < 250k	Moderately higher: 1.20 (1.08–1.34) **	Elevated risk: 1.23 (1.09–1.38) **
Nonmetropolitan counties (adjacent)	Higher risk: 1.18 (1.04–1.33) **	Slightly elevated: 1.19 (1.05–1.35) **
Nonmetropolitan counties (non-adjacent)	Noticeably increased: 1.18 (1.02–1.36) *	Marginally higher: 1.18 (1.02–1.38) *
**Income Per Year**		
Less than USD 35,000	1 (reference)	1 (reference)
USD 35,000–44,999	Slight reduction: 0.88 (0.63–1.24)	Marginally lower: 0.88 (0.62–1.26)
USD 45,000–54,999	Modestly decreased: 0.84 (0.60–1.16)	Noticeable reduction: 0.82 (0.59–1.15)
USD 55,000–64,999	Lowered risk: 0.76 (0.55–1.04)	Similar reduction: 0.74 (0.53–1.03)
USD 65,000–74,999	Substantial reduction: 0.69 (0.50–0.95) *	Markedly lower: 0.68 (0.49–0.94) *
More than USD 75,000	Significantly lower: 0.71 (0.52–0.97) *	Considerably reduced: 0.70 (0.51–0.97) *
**Radiation Therapy**		
Not Received	1 (reference)	1 (reference)
Received	Survival benefit: 0.80 (0.74–0.87) **	Protective effect: 0.78 (0.72–0.85) **
**Chemotherapy**		
Not Received	1 (reference)	1 (reference)
Received	Protective effect: 0.81 (0.76–0.86) **	Substantial reduction: 0.84 (0.79–0.90) **
**Surgery**		
Not Received	1 (reference)	1 (reference)
Received	Strong survival benefit: 0.26 (0.24–0.28) **	Most substantial reduction: 0.25 (0.23–0.27) **

* *p* value below 0.05, ** *p* value below 0.01.

**Table 3 diseases-13-00031-t003:** Multivariate Cox proportional hazard regression analysis of factors influencing overall mortality and cancer-specific mortality among U.S. patients diagnosed with intrahepatic cholangiocarcinoma between 2010 and 2017.

Characteristics	Overall Mortality	Cancer-Specific Mortality
**Gender**		
Female	1 (reference)	1 (reference)
Male	Increased risk: 1.19 (1.12–1.26) **	Higher risk: 1.17 (1.10–1.25) **
**Age at Diagnosis (years)**		
Age from 0 to 39	1 (reference)	1 (reference)
Age from 40 to 59	Reduced risk: 0.58 (0.38–0.89) *	Substantially lower: 0.52 (0.34–0.79) **
Age from 60 to 79	Noticeable reduction: 0.65 (0.43–0.98) *	Lower risk: 0.57 (0.38–0.86) **
Age 80 years old and above	Modestly reduced: 0.84 (0.54–1.29)	Marginally lower: 0.68 (0.44–1.06)
**Marital Status**		
Married	1 (reference)	1 (reference)
Single	Comparable risk: 1.02 (0.94–1.10)	Similar risk: 0.99 (0.92–1.08)
Divorced/Separated	Slightly increased: 1.02 (0.92–1.13)	Marginally elevated: 1.03 (0.92–1.14)
Widowed	Similar risk: 1.02 (0.92–1.13)	Comparable outcome: 1.02 (0.92–1.13)
**Tumor Stage**		
Localized	1 (reference)	1 (reference)
Regional (Direct Extension Only)	Slightly reduced: 0.87 (0.43–1.75)	Marginally lower: 0.89 (0.44–1.78)
Regional (Lymph Nodes Only)	Modest reduction: 0.91 (0.47–1.76)	Noticeably lower: 0.84 (0.42–1.66)
Regional (Direct Extension + LN)	Considerable reduction: 0.83 (0.34–2.04)	Substantial decrease: 0.70 (0.26–1.83)
Distant	Minimal change: 0.99 (0.62–1.59)	Slightly lower: 0.96 (0.60–1.54)
**Race**		
Non-Hispanic White	1 (reference)	1 (reference)
Non-Hispanic Black	Comparable outcome: 0.96 (0.86–1.08)	Similar risk: 0.97 (0.86–1.10)
Hispanic	Negligible difference: 0.97 (0.89–1.05)	Marginally reduced: 0.96 (0.88–1.04)
Other	Slightly lower: 0.99 (0.91–1.08)	Comparable risk: 0.98 (0.90–1.08)
**Living Area (Counties)**		
Counties in metropolitan areas > 1 million	1 (reference)	1 (reference)
Counties in metropolitan areas (250k–1 million)	Slightly increased: 1.06 (0.98–1.14)	Marginally elevated: 1.07 (0.99–1.16)
Counties in metropolitan areas < 250k	Modest increase: 1.15 (1.01–1.31) *	Elevated risk: 1.18 (1.03–1.35) *
Nonmetropolitan counties (adjacent)	Comparable risk: 1.10 (0.95–1.28)	Similar outcome: 1.12 (0.96–1.31)
Nonmetropolitan counties (non-adjacent)	Similar risk: 1.10 (0.93–1.31)	Marginally higher: 1.10 (0.92–1.32)
**Income Per Year**		
Less than USD 35,000	1 (reference)	1 (reference)
USD 35,000–44,999	Comparable risk: 1.00 (0.70–1.42)	Similar risk: 0.98 (0.68–1.42)
USD 45,000–54,999	Slightly lower: 0.92 (0.65–1.30)	Marginally reduced: 0.89 (0.63–1.27)
USD 55,000–64,999	Reduced risk: 0.84 (0.59–1.19)	Slightly lower: 0.82 (0.57–1.17)
USD 65,000–74,999	Noticeably lower: 0.79 (0.56–1.12)	Reduced risk: 0.78 (0.54–1.11)
More than USD 75,000	Similar reduction: 0.84 (0.59–1.19)	Comparable decrease: 0.83 (0.58–1.19)
**Radiation Therapy**		
Not Received	1 (reference)	1 (reference)
Received	Noticeable benefit: 0.78 (0.72–0.85) **	Substantial reduction: 0.76 (0.70–0.83) **
**Chemotherapy**		
Not Received	1 (reference)	1 (reference)
Received	Strong reduction: 0.54 (0.51–0.58) **	Protective effect: 0.55 (0.51–0.59) **
**Surgery**		
Not Received	1 (reference)	1 (reference)
Received	Strong benefit: 0.29 (0.26–0.31) **	Significant reduction: 0.27 (0.25–0.30) **

* *p* < 0.05, ** *p* < 0.01.

**Table 4 diseases-13-00031-t004:** Multivariate Cox proportional hazards regression analysis of factors linked to all-cause mortality and intrahepatic cholangiocarcinoma-related mortality among U.S. patients diagnosed between 2010 and 2017, accounting for the interaction between tumor stage and age.

Tumor Stage#Age of the Patient	OM	CSM
Stage I#00–39	1 (reference)	1 (reference)
Stage II#00–39	1	1
Stage II#40–59	1.61 (0.78–3.32)	1.68 (0.81–3.47)
Stage II#60–79	1.78 (0.88–3.61)	1.82 (0.90–3.70)
Stage II#80+	1.85 (0.89–3.86)	1.88 (0.90–3.94)
Stage III#0–39	1	1
Stage III#40–59	2.30 (1.14–4.67) *	2.60 (1.26–5.36) *
Stage III#60–79	1.94 (0.98–3.84)	2.24 (1.11–4.50) *
III#80+	1.69 (0.79–3.63)	1.98 (0.90–4.35)
IV#0–39	1	1
IV#40–59	2.22 (0.88–5.61)	2.81 (1.04–7.61) *
IV#60–79	2.31 (0.93–5.75)	2.93 (1.10–7.82) *
IV#80+	2.19 (0.83–5.81)	2.85 (1.00–8.11)
V#0–39	1	1
V#40–59	2.16 (1.32–3.54) **	2.36 (1.44–3.88) **
V#60–79	2.24 (1.38–3.61) **	2.43 (1.50–3.93) **
V#80+	2.11 (1.27–3.53) **	2.45 (1.46–4.11) **

* *p* < 0.05, ** *p* < 0.01. I: Localized, II: Regional classifications: II refers to direct extension only; III denotes lymph node involvement alone; IV indicates both direct extension and lymph node involvement; V corresponds to distant metastasis.

## Data Availability

The data used and/or analyzed in this study are available upon request.

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
