# Peer review of "Age and Tumor Stage Interplay in Intrahepatic Cholangiocarcinoma: Prognostic Factors, Mortality Trends, and Therapeutic Implications from a SEER-Based Analysis"

_diseases, 2025, doi:10.3390/diseases13020031_

Round 1

Reviewer 1 Report

Comments and Suggestions for Authors

In this work, the investigators re-analyzed data from 5083 iCCA patients from the Surveillance, Epidemiology, and End Results (SEER) database by employing statistical methods of calculating hazard, to highlight significant association between age, sex, marital status and tumor stage with overall mortality and cancer-specific mortality. While the results overall extend and/or support previous findings, it is interesting to note the worse outcome in young and middle aged people with advanced stage disease, compared to old age group. Given the current highlight on inflammation, and unhealthy lifestyle, can BMI/BRI, or lifestyle factors be included in the analysis, and can we see an association between age and lifestyle with OM, and CSM?

Author Response

Thank you for taking the time out of your busy schedule to review our manuscript. Important points noted. 

In this work, the investigators re-analyzed data from 5083 iCCA patients from the Surveillance, Epidemiology, and End Results (SEER) database by employing statistical methods of calculating hazard, to highlight significant association between age, sex, marital status and tumor stage with overall mortality and cancer-specific mortality.

Re: Thank you for recognizing the relevance of our study.  

While the results overall extend and/or support previous findings, it is interesting to note the worse outcome in young and middle aged people with advanced stage disease, compared to old age group. Given the current highlight on inflammation, and unhealthy lifestyle, can BMI/BRI, or lifestyle factors be included in the analysis, and can we see an association between age and lifestyle with OM, and CSM?

Re: The reviewer makes an excellent point. However, the SEER database used for this study does not provide information on BMI or lifestyle. And this was added as a limitation of the study. 

Reviewer 2 Report

Comments and Suggestions for Authors

Dear Authors 

I have the following comments

1. Line 114: 50.59% shall not be written as male dominance

2. Please provide data on co-morbidities such as DM, hypertension, COPD, CAD, obesity, cirrhosis, alcoholic liver disease etc

3. Please provide data on risk factors for ICC such as HBsAg, anti-HCV, intrahepatic parasites, recurrent cholangitis etc.

4. Please provide data on clinical presentation and duration of symptoms before their first presentation

5. Please describe the staging system used for ICC in teh manuscript

6. Data on chemotherapy and surgery shall not be there in table 1 becaus eit is talking about baselien information

7. Please provide the numbers and reasons for not giving chemotherapy or surgery or radiotherapy to study participants 

8. Table 2 & 3: Age is the biggest risk factor for death regardless of cause any where in the world. Hence, the death rate shall be adjusted for the age. Similalrly, the death rate shall also be adjusted for major co-morbidities such as DM, CAD etc

9. Moratlity shall also be corrected for the type treatment given for ICC

Author Response

Thank you for taking the time out of your busy schedule to review our manuscript. Important points noted. 

I have the following comments

  1. Line 114: 50.59% shall not be written as male dominance

Re: Thank you for this keen observation, the change was made. 

  1. Please provide data on co-morbidities such as DM, hypertension, COPD, CAD, obesity, cirrhosis, alcoholic liver disease etc

Re: The reviewer makes an excellent point. However, the SEER database used for this study does not provide information on BMI, lifestyle or comorbidities. And this was added as a limitation of the study. 

  1. Please provide data on risk factors for ICC such as HBsAg, anti-HCV, intrahepatic parasites, recurrent cholangitis etc.

Re: Data on risk factors of Cholangiocarcinoma was added in the introduction section. 

  1. Please provide data on clinical presentation and duration of symptoms before their first presentation

Re: Data on clinical presentation and symptoms before their first presentation was added to the introduction section. 

  1. Please describe the staging system used for ICC in the manuscript

Re: The staging system used in the database is called the SEER summary stage. The summary Stage is the most basic way of categorizing how far a cancer has spread from its point of origin. Historically, Summary Stage has also been called General Stage, California Stage, historic stage, and SEER Stage. The 2018 version of Summary Stage applies to every site and/or histology combination, including lymphomas and leukemias. Summary Stage uses all information available in the medical record; in other words, it is a combination of the most precise clinical and pathological documentation of the extent of disease. 

https://seer.cancer.gov/tools/ssm/

  1. Data on chemotherapy and surgery shall not be there in table 1 becaus eit is talking about baselien information

Re: The authors respectfully disagree with the suggestion, as we believe that the readers need to know the proportion of patients that received different treatment modalities. The authors have published similar studies in the past and this information was provided in Table 1. 

https://pmc.ncbi.nlm.nih.gov/articles/PMC11587058/ 

  1. Please provide the numbers and reasons for not giving chemotherapy or surgery or radiotherapy to study participants 

Re: Given the granularity of the database, this information is not provided. This is a central database where information of patients is de-identified. Patient information was not retrieved from chart review. 

  1. Table 2 & 3: Age is the biggest risk factor for death regardless of cause any where in the world. Hence, the death rate shall be adjusted for the age. Similalrly, the death rate shall also be adjusted for major co-morbidities such as DM, CAD etc

Re:  The authors used a univariate analysis followed by a  multivariate analysis that adjusts for all the covariates.

The Cox proportional hazards model assumes that the hazard ratios for covariates remain constant over time (the proportional hazards assumption). This is critical to ensure the validity of the results, particularly when analyzing time-to-event data. Proportionality testing is often performed using methods such as Schoenfeld residuals or time-dependent covariates to identify any violations. In univariate analysis, each covariate is assessed independently, which may overlook the influence of confounders. Multivariate analysis addresses this by simultaneously including multiple covariates, thereby accounting for potential confounding factors and interactions between variables. This step strengthens the model’s reliability by isolating the independent effect of each covariate while adjusting for others. Ensuring proportionality holds across all included covariates in both univariate and multivariate contexts is crucial for interpreting the hazard ratios as stable over time, reinforcing the validity of the findings.

Information on comorbidities is not provided in the database. 

  1. Mortality shall also be corrected for the type treatment given for ICC

To reiterate on the previous point. The authors used a univariate analysis followed by a  multivariate analysis that adjusts for all the covariates.

The Cox proportional hazards model assumes that the hazard ratios for covariates remain constant over time (the proportional hazards assumption). This is critical to ensure the validity of the results, particularly when analyzing time-to-event data. Proportionality testing is often performed using methods such as Schoenfeld residuals or time-dependent covariates to identify any violations. In univariate analysis, each covariate is assessed independently, which may overlook the influence of confounders. Multivariate analysis addresses this by simultaneously including multiple covariates, thereby accounting for potential confounding factors and interactions between variables. This step strengthens the model’s reliability by isolating the independent effect of each covariate while adjusting for others. Ensuring proportionality holds across all included covariates in both univariate and multivariate contexts is crucial for interpreting the hazard ratios as stable over time, reinforcing the validity of the findings.

Information on comorbidities is not provided in the database. 

Round 2

Reviewer 2 Report

Comments and Suggestions for Authors

Thanks for making changes in the manuscript